# Heterogeneous peer effects of college roommates on academic performance

Yi Cao [1,2], Tao Zhou [1,2] ✉ & Jian Gao [3,4,5,6] ✉

Understanding how student peers influence learning outcomes is crucial for effective education management in complex social systems. The complexities of peer selection and evolving peer relationships, however, pose challenges for identifying peer effects using static observational data. Here we use both null-model and regression approaches to examine peer effects using longitudinal data from 5,272 undergraduates, where roommate assignments are plausibly random upon enrollment and roommate relationships persist until graduation. Specifically, we construct a roommate null model by randomly shuffling students among dorm rooms and introduce an assimilation metric to quantify similarities in roommate academic performance. We find significantly larger assimilation in actual data than in the roommate null model, suggesting roommate peer effects, whereby roommates have more similar performance than expected by chance alone. Moreover, assimilation exhibits an overall increasing trend over time, suggesting that peer effects become stronger the longer roommates live together. Our regression analysis further reveals the moderating role of peer heterogeneity. In particular, when roommates perform similarly, the positive relationship between a student's future performance and their roommates' average prior performance is more pronounced, and their ordinal rank in the dorm room has an independent effect. Our findings contribute to understanding the role of college roommates in influencing student academic performance.

Peer effects, or peer influence[1–5], have long been studied in the literature on social contagions[6–11] and education[12–18]. Understanding the influence of student peers on social behavior and learning outcomes is crucial for effective education management[18–22], as it can inform policy decisions on how to improve learning environments inside and outside the classroom[23–28]. Student peers can have both positive and negative effects, depending on their characteristics and behaviors[29,30]. For example, when surrounded by high-achieving peers, students may be motivated to improve their academic performance[31,32]. Meanwhile, some well-known examples of human behaviors adopted through social influence, such as smoking[33,34], substance abuse[35,36], and alcohol use[37–39], are often associated with negative student performance. Moreover, student peers may have indirect and lasting effects, for instance, on political ideology[40], persistence in STEM majors[41–45], occupational preferences[46], labor market outcomes[47–49], and earnings[50–53]. A thorough understanding of peer effects on learning outcomes can inform education management strategies, such as implementing behavioral interventions to mitigate the negative influence of disruptive peers[54,55]. Yet, using traditional methods and observational data to study peer effects causally is a challenge.

Dynamic educational and social environments make it difficult to separate peer influence from peer selection due to reverse causality,

[1]CompleX Lab, University of Electronic Science and Technology of China, Chengdu, China. [2]Big Data Research Center, University of Electronic Science and Technology of China, Chengdu, China. [3]Center for Science of Science and Innovation, Northwestern University, Evanston, IL, USA. [4]Kellogg School of Management, Northwestern University, Evanston, IL, USA. [5]Northwestern Institute on Complex Systems, Northwestern University, Evanston, IL, USA. [6]Faculty of Social Sciences, The University of Hong Kong, Hong Kong SAR, China. ✉e-mail: zhutou@ustc.edu; jian.gao1@kellogg.northwestern.edu

confounding factors, and complex mechanisms[1–3,56]. In particular, similarities in academic performance among student peers may be due to homophily (i.e., the selection of peers based on academic performance similarity) rather than the influence of peers[57–59]. Unlike open and evolving educational environments such as classrooms[23–26], dormitories in universities provide a close-knit living environment for students to interact and potentially learn from each other[60,61]. While dorm rooms may not be the primary learning place like classrooms and libraries, they offer a highly interpersonal and spillover environment for a small group of stable student peers. In contrast to Western universities, in which freshman students usually have the flexibility to choose dormitories and suite-mates according to their lifestyle and personal preferences, most Chinese universities randomly assign students to dorm rooms[61–63]. There, a typical 4-person dorm room contains four beds and some public areas, providing a more interactive environment than a Western dorm suite containing four separate bedrooms (Supplementary Fig. 1).

Research on student peer effects, on the one hand, has primarily relied on static observational data of campus behaviors and performance metrics[11,64]. This reliance stems from various factors, such as the high cost and impracticality of conducting large-scale field experiments in learning environments, the dynamic nature of peer relationships[65], and the scarcity of longitudinal data on student learning outcomes[66–70]. The close-knit dormitory environment of Chinese universities, however, provides a unique opportunity to observe a stable group of student peers and track their academic performance over time[61,63]. On the other hand, while regression models are widely employed in studying peer effects within the social sciences, methodologies from other disciplines may help expand the functional form in which peer effects can be estimated[64]. Particularly, null models are well suited for studying nontrivial features of complex systems by providing an unbiased random structure of other features[71–73]. Null-model approaches have been applied to test causal effects in complex social systems[74–76]. For instance, in the social network literature, randomizations are used to study the impact of network interventions on social relationships[77]. Utilizing a null model to test whether roommates exhibit similar performance could offer a promising approach to identifying peer effects and quantifying their magnitude, facilitating comparisons across diverse datasets.

One advantage of regression models is their capability to address the issue of inverse causality by utilizing longitudinal data and controlling for confounding factors[68,78]. For example, a student's future performance may be influenced not only by the average prior performance of roommates but also by their own prior performance. Additionally, the composition of roommates may have independent effects[79]. Yet, it remains relatively less explored whether the heterogeneity in performance among roommates provides a ladder for the student to catch up with high-achieving roommates or hamper their motivation due to the inconsistent signal from roommates or the negative impact of disruptive roommates[29,30]. Moreover, dorm rooms provide an interactive yet local environment where a student's ordinal rank in the dorm room, conditional on academic performance, may independently affect learning outcomes[80,81]. Therefore, a more comprehensive understanding of the factors contributing to roommate peer effects may help inform education policy and student management strategies, such as designing interventions for dormitories that effectively leverage the influence of high-achieving peers in improving student performance.

In this study, we quantify roommate peer effects using both null models and regression approaches to analyze a longitudinal dataset of student accommodation and academic performance. Sourced from a public research-intensive university in China, our data covers 5,272 undergraduate students residing in 4-person dorm rooms following the random assignment of roommates (see "Methods"). The initialization is plausibly random since the roommate assignment takes into account neither students' academic performance before college admission nor their personal preferences, and there is no significant reassignment later (see Supplementary Information Section 1.2 for details). Here, we demonstrate the presence of roommate peer effects by showing that roommates with similar performance are more likely to be observed in the actual data than expected by chance alone. We then measure the size of roommate peer effects by developing an assimilation metric of academic performance and contrasting its value in the actual data with that in the roommate null model that we construct by randomly shuffling students among dorm rooms while retaining their controlled characteristics. Further, we use regression models to examine factors influencing roommate peer effects and explore the role of peer heterogeneity in moderating the effects.

## Results
### Tier combinations within a dorm room
We start by studying the roommate composition of a typical 4-person dorm room in terms of their academic performance. For comparisons across student cohorts (i.e., those who were admitted by the university in the same year), majors, and semesters, we transform each student's grade point average (GPA) in a semester into the GPA percentile $R$ among students in the same cohort and major, where $R = 0$ and $R = 1$ correspond to the lowest and highest academic performance, respectively. We then divide students into equal-sized tiers based on their GPA percentiles, where those with better performance are in larger tiers. For instance, under the 4-tier classification, students with $R = 0.3$ (i.e., GPA is above 30% of students) and $R = 0.9$ (i.e., GPA is above 90% of students) are in Tier 2 and Tier 4, respectively. Accordingly, each dorm room has a tier combination without particular order. For example, 3444 (i.e., one student is in Tier 3, and the other three are in Tier 4) is identical to 4344 and 4434. Here we use the one in ascending order of tier numbers to delegate all identical ones. Under the 2-tier classification, there are five unique tier combinations (1111, 1112, 1122, 1222, and 2222). The numbers are 15 and 35 under 3-tier and 4-tier classifications, respectively (Fig. 1a; see Supplementary Information Section 2.1 for details).

Given a tier for classification, the probability $P_a$ of observing a combination in the actual data can be calculated by the fraction of dorm rooms with the combination. The actual probabilities $P_a$ of observing different combinations (i.e., the frequency of observations), however, shouldn't be directly compared. This is because their theoretical probabilities $P_t$ are not always the same even when the tier numbers of roommates are independent of each other, i.e., there is no roommate peer effect (see Supplementary Table 1 and Supplementary Information Section 2.1). To give a simple example: under the 2-tier classification, the theoretical probability $P_t$ of combination 1112 is $C_4^1 \left(\frac{1}{2}\right)^3 \left(\frac{1}{2}\right) = \frac{1}{4}$, which is four times as big as that of combination 1111, namely, $\left(\frac{1}{2}\right)^4 = \frac{1}{16}$. This leads to the difficulty of assessing, by the value of $P_a$, whether a combination is over-represented or under-represented in the actual data. To address this challenge, we calculate the relative ratio $\mathbb{E}$ for a combination by comparing the actual probability with its theoretical probability:

$$\mathbb{E} = \frac{P_a - P_t}{P_t}, \tag{1}$$

where $P_a$ and $P_t$ are the actual and theoretical probability of the same combination, respectively. A positive (negative) value of $\mathbb{E}$ suggests that the combination is more (less) likely to be observed in data than expected by chance alone (see Supplementary Information Section 2.2).

We analyze the student accommodation and academic performance data under 2-tier, 3-tier, and 4-tier classifications and calculate

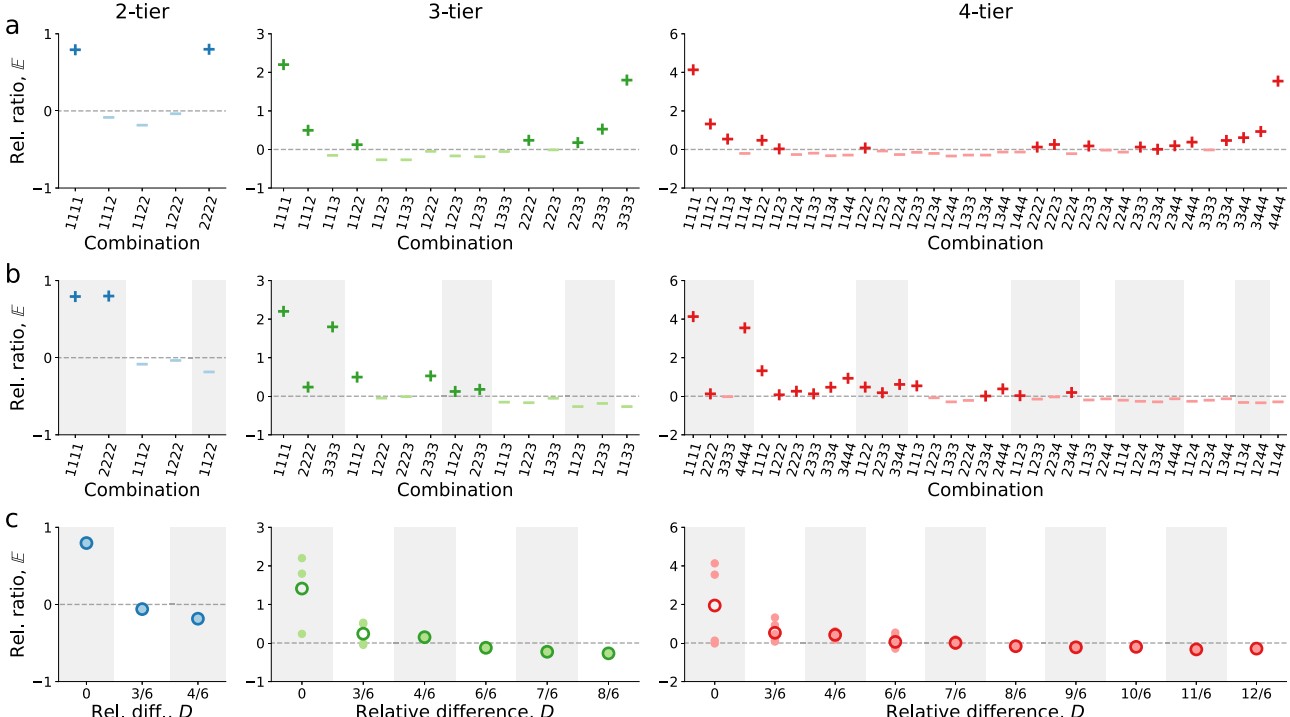

**Fig. 1 | The combinations of roommate tiers in a 4-person dorm room. a** The relative ratio $\mathbb{E}$ of each combination under the 2-tier, 3-tier, and 4-tier classification of GPA, respectively. The x-axis shows all unique combinations in ascending order of tier numbers under a tier classification, and the y-axis shows the relative ratio $\mathbb{E}$ that compares the actual frequency of a combination with its theoretical value. The horizontal dashed line marks 0. Positive and negative $\mathbb{E}$ is marked by '+' and '-', respectively. **b** Combinations in ascending order of the relative difference $D$, which measures the average pairwise difference between tier numbers of a combination. The staggered shade marks a group of combinations with the same $D$. **c** The negative relationship between the relative ratio $\mathbb{E}$ and the relative difference $D$ based on the actual data. Data points show the $\mathbb{E}$ for each combination, and the hollow circle shows the mean $\mathbb{E}$ for each group with the same $D$.

the relative ratio $\mathbb{E}$ for each combination (Fig. 1a). We find that $\mathbb{E}$ of different combinations vary substantially and $\mathbb{E}$ of some combinations deviates significantly from 0 according to the results of statistical tests (see "Methods" and Supplementary Information Section 3.2 for details). For example, under the 2-tier classification, $\mathbb{E}$ of combinations 1111 and 2222 is significantly above 0 and $\mathbb{E}$ of combinations 1112 and 1122 are significantly below 0 ($P$ value < 0.001; see Supplementary Table 2 for the statistical testing results for each combination). More notably, we find that combinations with the same or nearby tier numbers (e.g., 1111 and 1112) tend to have larger $\mathbb{E}$ and those with distant tier numbers (e.g., 1122) have smaller $\mathbb{E}$, prompting us to study the relationship between a combination's tier heterogeneity and its $\mathbb{E}$. Specifically, we first calculate the relative difference $D$ in the tier numbers for each combination:

$$D = \frac{1}{6}\sum_{u \neq v}|l_u - l_v|, 1 \le u < v \le 4, \quad (2)$$

where $l_u$ and $l_v$ is the tier number of roommates $u$ and $v$, respectively. A smaller $D$ indicates that roommates have closer tier numbers and thus a smaller difference in their academic performance. We then group combinations with the same $D$ and arrange them in ascending order of $D$. We find that combinations with positive and negative $\mathbb{E}$ are overall separated (Fig. 1b), where those with a smaller $D$ tend to have positive $\mathbb{E}$ (i.e., over-represented in the actual data) and those with a larger $D$ tend to have negative $\mathbb{E}$ (i.e., underrepresented in the actual data). Inspired by this observation, we calculate the mean value of $\mathbb{E}$ for each group with the same $D$, finding a negative relationship between $D$ and $\mathbb{E}$ (Fig. 1c). These results demonstrate that roommates tend to have more similar academic performance than random chance, suggesting the presence of roommate peer effects.

### Assimilation of roommate academic performance

We generalize the tier combination analysis to the most granular tier for classification by directly dealing with the GPA percentile $R \in [0, 1]$ (hereafter GPA for short). Specifically, similar to calculating the relative difference $D$ in the tier combination for each dorm room, we develop an assimilation metric $A$ to quantify the extent to which the GPAs of roommates differ from each other. Formally, the assimilation metric $A$ for a 4-person dorm room is calculated by

$$A = 1 - \frac{1}{4}\sum_{u \neq v}|R_u - R_v|, 1 \le u < v \le 4, \quad (3)$$

where $R_u$ and $R_v$ are the GPAs of roommates $u$ and $v$, respectively. The assimilation $A$ of a dorm room is between 0 and 1, with a larger value indicating that roommates have more similar academic performance. If there is no roommate peer effect, each roommate's GPA should be independent and identically distributed (i.i.d.), and the theoretical assimilation $A$ of all dorm rooms has a mean value of 0.5 (see Supplementary Information Section 4.1 for detailed explanations).

Inspired by permutation tests, often referred to as the "quadratic assignment procedure" in social network studies[74,75], we perform a statistical hypothesis test to check whether the assimilation of dorm rooms in the actual data deviates significantly from its theoretical value. Specifically, we proxy theoretical assimilation via null-model assimilation that is calculated based on a roommate null model and compare it with actual assimilation. An appropriate null model of a complex system satisfies a collection of constraints and offers a baseline to examine whether displayed features of interest are truly nontrivial[71–73]. We start with the actual roommate configuration and randomly shuffle students between dorm rooms while preserving their compositions of cohort, gender, and major. By repeating this process,

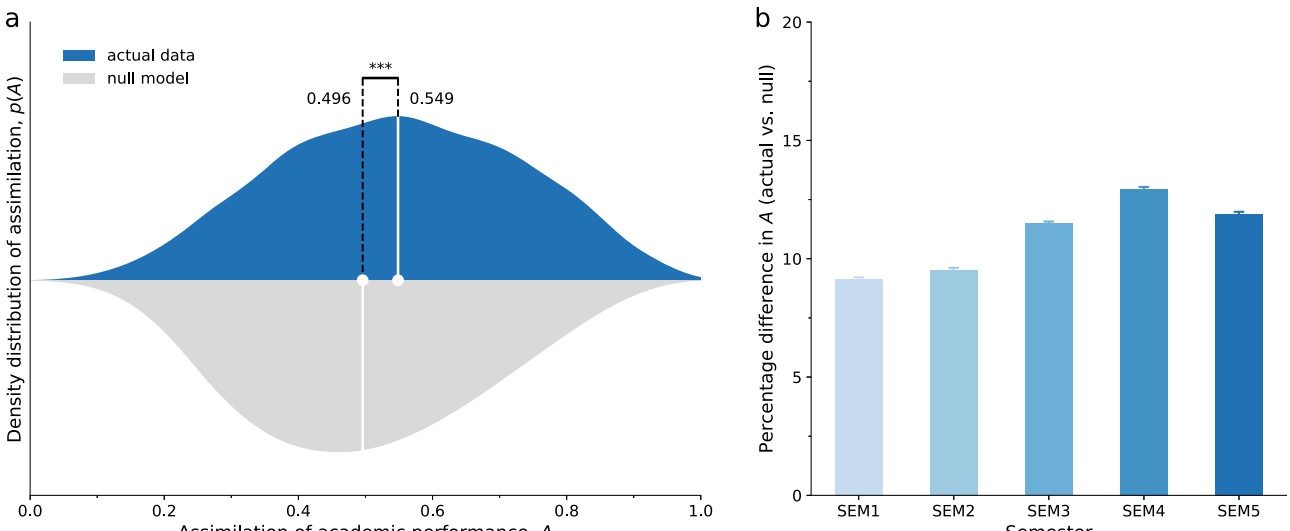

**Fig. 2 | The assimilation of roommate academic performance. a** The density distribution $p(A)$ of assimilation $A$ for all dorm rooms. Larger assimilation means roommates have more similar academic performance. The upper half (in blue) shows the actual assimilation and the lower half (in gray) shows the null-model assimilation. Vertical dashed lines mark the statistically different means of the two assimilation distributions based on a Student's $t$-test (***$P$ value < 0.001). The mean actual assimilation is 10.7% larger than the mean null-model assimilation, which is close to its theoretical value 0.5. The plot is based on the data from all five semesters. **b** The overall increasing trend in the actual assimilation from semester 1 to semester 5. The y-axis shows the percentage difference between the mean actual assimilation and the mean null-model assimilation. Error bars represent standard errors clustered for 100 times of independent implementations.

we construct a plausible roommate null model that consists of 1000 independent implementations (see Supplementary Information Section 3.1 for details). We find that the mean of actual assimilation (0.549) of all dorm rooms is 10.7% larger than that of null-model assimilation (0.496; Fig. 2a). A Student's $t$-test confirms that the two assimilation distributions have significantly different means ($P$ value < 0.001; see Supplementary Information Section 4.2 for details). These results suggest that roommate assimilation in academic performance is greater than expected by chance alone, demonstrating significant roommate peer effects.

The extent to which the mean of actual assimilation is larger than that of null-model assimilation indicates the magnitude of roommate peer effects, allowing us to examine temporal trends over the five semesters. First, we find that roommate peer effects remain significant when measured using data from each semester (see Supplementary Information Section 4.1 for details). Second, we hypothesize that before the first semester (i.e., the first day of college), roommate peer effects should be 0 due to the plausible random assignment of roommates, where the actual assimilation should be close to the null-model assimilation. As roommates live together longer and establish stronger interactions with each other, the actual assimilation of roommate academic performance would become larger, and the magnitude of roommate peer effects would become bigger. To test this hypothesis, for each semester, we calculate the percentage difference in the means of the actual assimilation and the null-model assimilation that is a proxy of roommate peer effects before the first semester (see Supplementary Information Section 4.1 for details). We find that the percentage difference exhibits an overall increasing trend over time (Fig. 2b), which supports the hypothesis that, as roommates live together longer, the magnitude of roommate peer effects on academic performance becomes larger. These results are robust when we use an alternative way to estimate the magnitude of roommate peer effects, where we calculate the share of dorm rooms with larger-than-null-model assimilation (see Supplementary Information Section 4.3 for details). Moreover, our further analysis shows that female and male students have similar assimilation, suggesting no significant gender differences (see Supplementary Information Section 4.4 for details).

## The effects of heterogeneous peers

The increasing assimilation of roommates in their academic performance raises a question about how a student's future performance is impacted by their roommates' prior performance, especially when there is substantial peer heterogeneity in performance, e.g., there are both high-achieving and underachieving roommates. To answer this question, we employ regression models to perform a Granger causality type of statistical analysis. Specifically, we first examine the relationship between a student's post-GPA (GPA_Post; e.g., their own GPA in the second semester) and the average prior GPA of their roommates (RM_Avg; e.g., their roommate's average GPA in the first semester) by calculating pairwise correlations for all consecutive semesters and dorm rooms. We find that dorm rooms tend to occupy the diagonal of the "GPA_Post – RM_Avg" plane (Fig. 3a), suggesting that a student's post-GPA is positively associated with the average prior GPA of their roommates. We then use an ordinary least squares (OLS) model to study the relationship between GPA_Post and RM_Avg (see "Methods" for the empirical specification) and summarize the regression results in Table 1. We find that without controlling for the effects of other factors (see column (1) of Table 1), the average prior GPA of roommates has a significantly positive effect on a student's post-GPA (regression coefficient $b = 0.365$; $P$ value < 0.001; Fig. 3b).

Other factors may independently affect a student's post-GPA and confound its association with the average prior GPA of their roommates. Therefore, we add controls and fixed effects into the OLS model (see "Methods"). The regression results shown in Table 1 convey several findings. First, a student's prior GPA has the strongest effect on their post-GPA ($b = 0.801$, which is 16 times as large as $b = 0.050$ for roommate average prior GPA; see columns (2) of Table 1), suggesting a significant path dependence on academic achievement. Second, the positive effect of roommate average prior GPA on a student's post-GPA remains significant with controlling the student's prior GPA, gender, cohort, major, and semester ($P$ value < 0.01; see column (2) of Table 1 and Fig. 3c). Notably, female students perform better than male students on average (see Supplementary Information Section 4.4 for details). Third, the differences in roommate prior GPAs (RM_Diff) have no significant effect ($P$ value > 0.1; see columns (3) and (4) of Table 1), but it significantly moderates the relationship between roommate

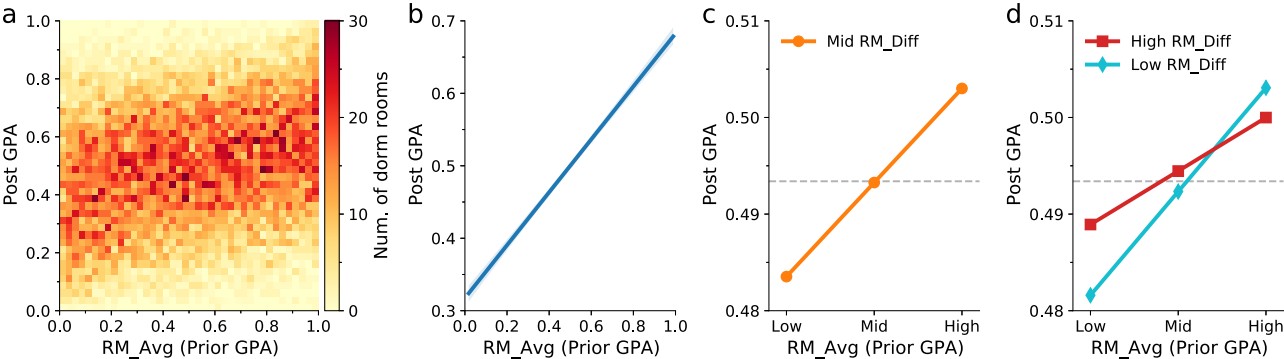

**Fig. 3 | The effects of roommate academic performance. a** The two-dimensional histogram shows the distributions of dorm rooms on the "GPA_Post – RM_Avg" plane. The y-axis shows the student's post-GPA (GPA_Post), and the x-axis shows the average prior GPA of roommates (RM_Avg). It shows a positive correlation between GPA_Post and RM_Avg (Pearson's $r = 0.244$; $P$ value < 0.001). **b** The regression plot for the relationship between GPA_Post and RM_Avg (center line) with the 95% confidence intervals (error bands), where the model includes no controls. **c** The plot for the relationship between GPA_Post and RM_Avg, where the model includes

controls and fixed effects (see Table 1 for details). The "Low" and "High" on the x-axis represent 1 standard deviation (SD) below and above the mean ("Mid") of RM_Avg, respectively. The horizontal dashed line marks the regression constant. **d** The plot for the moderating effects of peer heterogeneity. The relationship between GPA_Post and RM_Avg is moderated by the differences in roommate prior GPAs (RM_Diff). The "Low" and "High" in the legend represent 1 SD below and above the mean ("Mid") of RM_Diff, respectively. The horizontal dashed line marks the regression constant.

average prior GPA and post-GPA (see column (5) of Table 1 and Fig. 3d) such that their positive relationship is more pronounced (slope $b = 0.055$; 95% CI = [0.040, 0.070]) when RM_Diff is high (i.e., 1 SD above its mean) and less pronounced (slope $b = 0.028$; 95% CI = [−0.001, 0.057]) when RM_Diff is low (i.e., 1 SD below its mean; see Supplementary Information Section 5.1 for detailed results of a simple slope test). The result also shows that high post-GPA is associated with large differences in the roommate's prior GPA when the roommate's average prior GPA is low (see the red line on the lower left of Fig. 3d).

While the regression results suggest that roommate peer effects are significant, it is worth noting that the effect size appears to be modest. Specifically, a 100-point increase in roommate average prior GPA is associated with a 5-point increase in post-GPA ($b = 0.050$; see column (4) of Table 1). The effect is about 6% as large as the effect of a 100-point increase in prior GPA ($b = 0.801$), and it is about 10% of the average post-GPA. The magnitude is at a similar scale as reported by prior studies for various environments (e.g., dormitories and classrooms) and cultures (e.g., Western universities; see Supplementary Information Section 5.1 for details). To demonstrate its significance, we perform a falsification test by running the same OLS regression on the roommate null model, finding that the reported results are nontrivial (see Supplementary Information Section 5.3 for details). Together, these regression results suggest that a student's performance is impacted not only by the average performance of roommates but also by their heterogeneity in academic performance.

## The effects of in-dorm ordinal rank

Dorm rooms provide a highly interpersonal yet local environment, where competitive dynamics between roommates may affect their academic performance. Conditional on absolute academic performance, the ordinal rank of a student in their dorm room could have an independent effect on future achievement[80,81]. For instance, when a student's ordinal rank is consistently low across all semesters, even if their absolute performance is high (e.g., the student has a GPA $R = 0.9$ and their roommates all have $R > 0.9$), they may still feel discouraged and less motivated, leading to fewer interactions with others and a potential decline in performance (see Supplementary Information Section 5.2 for explanations). This motivates us to study how a student's in-dorm ordinal rank (OR_InDorm, with 1 being the highest and 4 being the lowest according to their prior performance; i.e., the number of better-achieving roommates including themself) affects their post-GPA. Specifically, we employ an OLS model that not only controls the

student's prior GPA, their roommate's average prior GPA, and differences in prior GPAs, gender, and semester but also includes the fixed effects of cohort and major (see "Methods" for the empirical specification). We find that ordinal rank has a significantly positive effect on post-GPA ($P$ value < 0.05; see columns (1) and (2) of Table 2 and Fig. 4a), suggesting that the number of better-achieving roommates in the dorm room predicts a student's better academic performance in the future.

Through regression, we further examine whether the positive relationship between ordinal rank and post-GPA is moderated by other factors. We find that neither the interaction term of ordinal rank and own prior GPA nor the interaction term of ordinal rank and average roommate's prior GPA is significant ($P$ value > 0.1; see columns (3) and (4) of Table 2). Yet, the interaction term of ordinal rank and differences in roommate prior GPA (RM_Diff) is significantly negative ($P$ value < 0.05; see columns (5) of Table 2). Specifically, the effect of ordinal rank on post-GPA is more pronounced (slope $b = 0.007$; 95% CI = [0.002, 0.012]) when RM_Diff is low (Fig. 4b), while the effect is not significant (slope $b = −0.000$; 95% CI = [−0.007, 0.007]) when RM_Diff is high (see Supplementary Information Section 5.2 for detailed results of a simple slope test). The result also shows that high post-GPA is associated with large differences in roommate prior GPA when ordinal rank is low (see the red line on the lower left of Fig. 4b). Although the effect size is modest, our falsification test on the roommate null model demonstrates that the results are nontrivial and significant (see Supplementary Information Section 5.3 for details). Taken together, these results suggest that roommate peer effects tend to disproportionately benefit underachieving students with homogeneous roommates (i.e., those who have similar performance) and high-achieving students with heterogeneous peers (i.e., those who have widely varied performance).

## Discussion

We quantified roommate peer effects on academic performance by applying both null-model and regression approaches to analyze a longitudinal dataset of student accommodation and academic performance, where roommate assignments are plausibly random upon enrollment and roommate relationships persist until graduation. We found evidence showing that roommates have a direct influence on a student's performance, with some heterogeneity in the variation among the roommates and the baseline achievement of the student. Specifically, by constructing a roommate null model and calculating an assimilation metric, we showed that roommates have more similar performance than expected by chance alone. Moreover, the average

**Table 1 | Summary of regression results on the relationship between a student's post-GPA and the average prior GPA of their roommates**

| Variables | Dependent variable: GPA_Post | | | | |
|---|---|---|---|---|---|
| | **(1)** | **(2)** | **(3)** | **(4)** | **(5)** |
| RM_Avg | 0.365*** | 0.050*** | | 0.050*** | 0.042*** |
| | (0.011) | (0.007) | | (0.007) | (0.009) |
| RM_Diff | | | 0.009 | 0.007 | 0.007 |
| | | | (0.009) | (0.009) | (0.009) |
| Interaction: RM_Avg × RM_Diff | | | | | −0.089* |
| | | | | | (0.052) |
| GPA_Prior | | 0.801*** | 0.809*** | 0.801*** | 0.801*** |
| | | (0.004) | (0.004) | (0.004) | (0.004) |
| D (gender) | | 0.039*** | 0.043*** | 0.039*** | 0.039*** |
| | | (0.004) | (0.004) | (0.004) | (0.004) |
| D (semester) | No | Yes | Yes | Yes | Yes |
| Major FE | No | Yes | Yes | Yes | Yes |
| Cohort FE | No | Yes | Yes | Yes | Yes |
| Observations | 15,680 | 15,680 | 15,680 | 15,680 | 15,680 |
| Adjust $R^2$ | 0.059 | 0.668 | 0.667 | 0.668 | 0.668 |
| RMSE | 0.284 | 0.169 | 0.169 | 0.169 | 0.169 |

Notes: Independent variables are mean-centered before being included in the regression models except for gender, semester, major, and cohort dummies. Females are in the treatment group. D (·) represents dummy variable, and FE represents fixed effects. Robust standard errors are reported in parentheses. Significant levels: *$P < 0.1$, **$P < 0.05$, ***$P < 0.01$.

**Table 2 | Summary of regression results on the relationship between a student's post-GPA and their in-dorm ordinal rank according to prior GPA**

| Variables | Dependent variable: GPA_Post | | | | |
|---|---|---|---|---|---|
| | **(1)** | **(2)** | **(3)** | **(4)** | **(5)** |
| OR_InDorm | 0.011*** | 0.006** | 0.006** | 0.006** | 0.003 |
| | (0.002) | (0.002) | (0.002) | (0.002) | (0.003) |
| RM_Avg | | 0.030*** | 0.030*** | 0.030*** | 0.035*** |
| | | (0.010) | (0.010) | (0.010) | (0.010) |
| RM_Diff | | 0.007 | 0.007 | 0.008 | 0.007 |
| | | (0.009) | (0.009) | (0.009) | (0.009) |
| Interaction 1: OR_InDorm × GPA_Prior | | −0.001 | | | |
| | | (0.004) | | | |
| Interaction 2: OR_InDorm × RM_Avg | | | | 0.004 | |
| | | | | (0.006) | |
| Interaction 3: OR_InDorm × RM_Diff | | | | | −0.022** |
| | | | | | (0.009) |
| GPA_Prior | 0.840*** | 0.820*** | 0.820*** | 0.820*** | 0.812*** |
| | (0.006) | (0.009) | (0.009) | (0.009) | (0.010) |
| D (gender) | 0.040*** | 0.039*** | 0.039*** | 0.039*** | 0.039*** |
| | (0.004) | (0.004) | (0.004) | (0.004) | (0.004) |
| D (semester) | Yes | Yes | Yes | Yes | Yes |
| Major FE | Yes | Yes | Yes | Yes | Yes |
| Cohort FE | Yes | Yes | Yes | Yes | Yes |
| Observations | 15,680 | 15,680 | 15,680 | 15,680 | 15,680 |
| Adjust $R^2$ | 0.668 | 0.669 | 0.669 | 0.669 | 0.669 |
| RMSE | 0.169 | 0.169 | 0.169 | 0.169 | 0.169 |

*Notes:* Independent variables are mean-centered before being included in the regression models except for gender, semester, major, and cohort dummies. Females are in the treatment group. D (·) represents dummy variable, and FE represents fixed effects. Robust standard errors are reported in parentheses. Significant levels: *$P < 0.1$, **$P < 0.05$, ***$P < 0.01$.

assimilation of roommate academic performance exhibits an overall increasing trend over time, suggesting that peer effects become stronger as roommates live together longer, get more familiar with each other, and establish stronger interactions that facilitate knowledge spillovers[61,65,82]. More specifically, the increase in assimilation is more pronounced in the third semester (Fig. 2b and Supplementary Fig. 8), which is consistent with previous literature showing that peer effects are strong and persistent when friendships last over a year[79,83], and it appears to be disrupted in the fifth semester, which may be because senior students have a higher chance of taking different elective courses and have more outside activities that might decrease the interactions between roommates[84].

Our regression analysis further unpacks roommate peer effects, especially along the dimension of peer heterogeneity. We found that a student's future performance is not only strongly predicted by their prior performance, suggesting a significant path dependence in academic development[85–87], but also impacted by their roommates' prior performance. Also, the positive relationship between a student's future performance and the average prior performance of roommates is moderated by peer heterogeneity such that it is more pronounced when roommates are similar. In particular, when living with roommates who have, on average low prior performance, a student benefits more if roommates are more different, suggesting the positive role of peer heterogeneity[88–90]. Moreover, ordinal rank in the dorm room has an independent effect since the number of better-achieving roommates is positively associated with future performance. Yet, peer heterogeneity moderates this relationship such that it is significant only when roommates are more similar. The magnitudes of peer effects assessed using regression may appear modest, but they are significant and in line with the literature. Together, these results paint a rich picture of roommate peer effects and suggest that the effective strategy for improving a student's performance may depend on their position in a high-dimensional space of ordinal rank, peer average performance, and peer heterogeneity.

While our work helps better understand roommate peer effects, the results should be interpreted in light of the limitations of the data and analysis. First, the longitudinal data were limited to two cohorts of Chinese undergraduates in one university. The extent to which these findings can be generalized to other student populations, universities, and countries should be further investigated where relevant data on student accommodation and academic outcomes are available. Second, the roommate assignments were plausibly random according to the administrative procedures. While providing some supporting evidence for this assumption (see Supplementary Information Section 1.2 for details), we lacked comprehensive data on student demographics, personal information, and pre-college academic performance to examine it directly. Third, the analysis relies on GPA percentiles normalized for each cohort and major, which allows for fair comparisons between disciplines but, at the same time, may lose more information in the data. A better normalization that preserves the distribution of GPAs, for example, would be an improvement. Fourth, factors outside of the dormitory environment may mediate the assimilation of roommates' academic performance, such as orderliness, classroom interactions, social networks, behavior patterns, and common external factors[16,17,65]. Unraveling the mechanisms underlying roommate peer effects (e.g., peer pressure and student identity[91]) was beyond the reach of this study but is desirable as future work.

In summary, we demonstrate the peer effect of college roommates and assess its magnitude by employing basic statistical methods to analyze new longitudinal data from a quasi-experiment. The university dorm room environment is ideal for identifying a group of frequently interacting and stable student peers whose learning outcomes can be easily tracked over time. The null model we use, which is

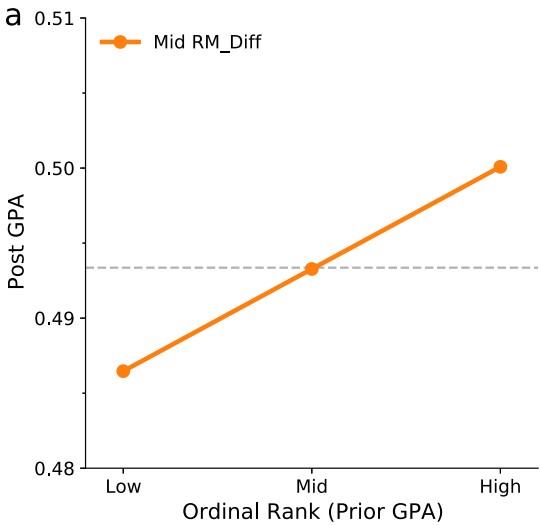
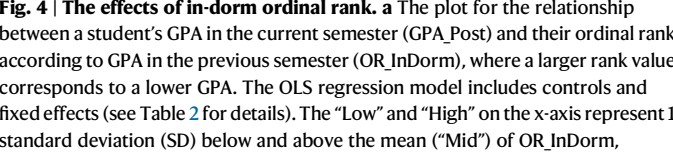
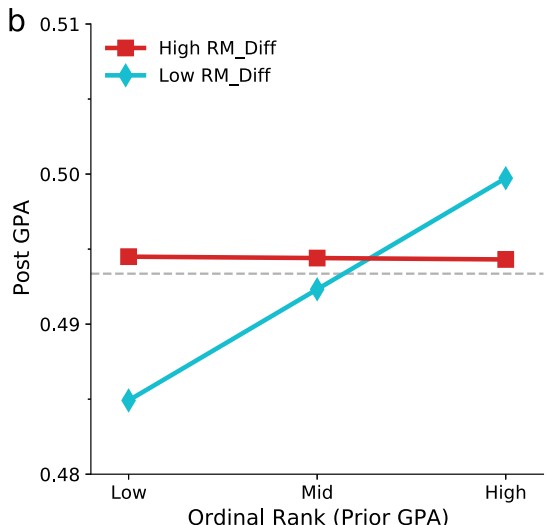

**Fig. 4 | The effects of in-dorm ordinal rank. a** The plot for the relationship between a student's GPA in the current semester (GPA_Post) and their ordinal rank according to GPA in the previous semester (OR_InDorm), where a larger rank value corresponds to a lower GPA. The OLS regression model includes controls and fixed effects (see Table 2 for details). The "Low" and "High" on the x-axis represent 1 standard deviation (SD) below and above the mean ("Mid") of OR_InDorm, respectively. The horizontal dashed line marks the regression constant. **b** The plot for the moderating effects of peer heterogeneity. The relationship between GPA_Post and OR_InDorm is moderated by the differences in roommate GPAs in the previous semester (RM_Diff). The "Low" and "High" in the legend represent 1 SD below and above the mean ("Mid") of RM_Diff, respectively. The horizontal dashed line marks the regression constant.

essentially permutation tests[75,76], does not assume linear relationships between variables and is flexible enough to be applied to study peer effects in other complex social systems. Also, effect sizes assessed by the null model can facilitate comparisons between different datasets. Moreover, the regression model allows us to address concerns about inverse causality and better understand peer effects. Particularly, the regression findings have potential policy implications for education and dormitory management. For example, by adjusting the composition of roommates, such as reducing peer heterogeneity for students with, on average, high-achieving roommates, dorm rooms may be engineered, to some extent, to enhance the positive influence of roommates in improving students' academic performance. Furthermore, our findings suggest the benefits of exposure to student role models and learning from peers in everyday life in addition to teachers in classrooms only.

## Methods
### Data
Chinese universities provide on-campus dormitories for almost all undergraduates, allowing us to observe a large-scale longitudinal sample of student roommates and relate it to their academic performance. From a public university in China, we collected the accommodation and academic performance data of 5,272 undergraduates, who lived in identical 4-person dorm rooms in the same or nearby dorm building on campus. Different from a dorm suite that contains four separate bedrooms, a 4-person dorm room is a single bedroom with four beds, where each student occupies one bed and shares public areas with roommates (see Supplementary Fig. 1 for an example layout). Per the university's student accommodation management regulations, newly admitted students were assigned to dorm rooms under the condition that those in the same administrative unit, major, or school live together as much as possible and there is no gender mix in dorm rooms or buildings. The process neither allowed students to choose roommates or rooms nor took into account their academic performance before admission, socioeconomic backgrounds, or personal preferences. Students were informed of their accommodation only when they moved in before the first semester. As a quasi-experiment, the administrative procedure resulted in a plausibly, if not

perfect, random assignment of roommates concerning their prior academic performance and personal information. Moreover, there was no significant individual selection later in the semesters. Once assigned together, roommates lived together until their graduation. Moving out or changing roommates was very rare on a few occasions (see Supplementary Information Section 1.2 for more details).

The dataset covers two cohorts of Chinese undergraduates who were admitted by the university in 2011 and 2012, respectively. For each student, we solicited information about their cohort, gender, major, and dorm room, based on which we determined roommate relationships. As a measure of academic performance, we collected the GPA data of these students for the first five successive semesters up to 2014 and further normalized it for each semester to a GPA percentile for students in the same cohort and major (see Supplementary Information Section 1.2 for details). The stable roommate relationship and the longitudinal academic performance data allowed us to study how a student is affected by roommates over time. All students were anonymized in the data collection and analysis process, and the dataset contains no identifiable information. This study was approved by the Institutional Review Board (IRB) at the University of Electronic Science and Technology of China (IRB No. 1061420210802005).

### Statistical hypothesis test
Given a tier of classification for students' GPA, following permutation tests[74–76], we perform a statistical test to examine whether the relative ratio $\mathbb{E}$ of each combination (e.g., 1111) in the actual data deviates significantly from its theoretical value 0. Specifically, we generate a roommate null model by implementing the random shuffling process and calculate the null-model relative ratio for each combination: $\widetilde{\mathbb{E}} = (P_n - P_t)/P_t$, where $P_n$ and $P_t$ is the null-model and theoretical frequency of the combination, respectively. By null-model construction, $P_n$ should approach $P_t$, and thus $\widetilde{\mathbb{E}}$ should be close to 0. For each combination, we compare the actual $\mathbb{E}$ with its null-model $\widetilde{\mathbb{E}}$. If $\mathbb{E}$ is significantly above 0, the probability of observing $\mathbb{E} \leq \widetilde{\mathbb{E}}$ in the actual data should be sufficiently small, e.g., less than 0.001. Accordingly, our null hypothesis (H0) is $\mathbb{E} \leq \widetilde{\mathbb{E}}$, and the alternative hypothesis (H1) is $\mathbb{E} > \widetilde{\mathbb{E}}$. To empirically test H0, we generate 1000 roommate null models (where each null model is an independent implementation

of the random shuffling process) and calculate $\widetilde{\mathbb{E}}$ under 2-tier, 3-tier, and 4-tier classifications, respectively. We find that $\mathbb{E}$ of some combinations is larger than $\widetilde{\mathbb{E}}$ for all 1000 roommate null models, allowing us to reject H0 and support H1 (i.e., $\mathbb{E}$ is significantly larger than 0 with a $P$ value < 0.001 in the one-sided statistical test; the combination is over-represented in the actual data). Similarly, we test whether $\mathbb{E}$ of a combination is significantly below 0. Under the 2-tier classification, for example, combinations with significantly positive $\mathbb{E}$ include 1111 and 2222 ($P$ value < 0.001) and those with significantly negative $\mathbb{E}$ include 1112 and 1122 ($P$ value < 0.001) as well as 1222 ($P$ value < 0.05; see Supplementary Table 2 for the statistical testing results for each combination under these tier classifications). Overall, we find that significantly positive combinations have the same or nearby tier numbers and significantly negative ones have distant tier numbers.

To perform a single statistical test for all combinations together given a tier of classification, we calculate the total relative ratio $\sum |\mathbb{E}|$ and $\sum \left| \widetilde{\mathbb{E}} \right|$ by summing up the absolute $\mathbb{E}$ and $\widetilde{\mathbb{E}}$ of each combination, respectively. As $\widetilde{\mathbb{E}}$ is close to 0, $\sum \left| \widetilde{\mathbb{E}} \right|$ should also be close to 0. If we assume $\sum |\mathbb{E}| \le \sum \left| \widetilde{\mathbb{E}} \right|$, it is naturally that $\sum |\mathbb{E}|$ is close to 0, yielding $\mathbb{E}$ to be close to 0. There, $\mathbb{E}$ and $\widetilde{\mathbb{E}}$ wouldn't have a significant difference because they are both close to 0. Thereby, to say $\mathbb{E}$ is significantly different from $\widetilde{\mathbb{E}}$, the probability of observing $\sum |\mathbb{E}| \le \sum \left| \widetilde{\mathbb{E}} \right|$ should be sufficiently small, e.g., less than 0.001. Accordingly, our null hypothesis (H0) is $\sum |\mathbb{E}| \le \sum \left| \widetilde{\mathbb{E}} \right|$, and the alternative hypothesis (H1) is $\sum |\mathbb{E}| > \sum \left| \widetilde{\mathbb{E}} \right|$. We find that, under 2-tier, 3-tier, and 4-tier classifications, $\sum |\mathbb{E}|$ is always larger than $\sum \left| \widetilde{\mathbb{E}} \right|$ for all 1000 roommate null models, allowing us to reject H0 and support H1 with a $P$ value < 0.001 (i.e., the overall $\mathbb{E}$ of all combinations is different from 0). Taken together, our hypothesis testing results suggest that $\mathbb{E}$ of some combinations in the actual data deviate significantly from 0, where those with nearby tier numbers are more likely to be observed and those with distant tier numbers are less likely to be observed than random chance, suggesting significant roommate peer effects (see Supplementary Information Section 3.2 for details).

### Regression model

We employ an ordinary least squares (OLS) model to study the relationship between a student's future performance (GPA_Post) and the average prior performance of their roommate (RM_Avg) and how this relationship is moderated by the differences in roommate prior performance (RM_Diff). The OLS model includes several controls on student demographics and prior performance. Specifically, the empirical specification is given by

$$G_i^{s+1} = b_0 + b_1 G_i^s + b_2 RA_i^s + b_3 RD_i^s + b_4 RA_i^s \times RD_i^s + b_5 D^{Ge} + b_6 D^{Ma} + b_7 D^{Co} + b_8 D^{Se} + \epsilon_i, \tag{4}$$

where $\epsilon_i$ is the error term for student $i$, and the semester index $s$ ranges from 1 to 4. The dependent variable $G_i^{s+1}$ is the student's GPA in semester $s+1$ (GPA_Post), and the independent variable of interest $G_i^s$ is the student's GPA in semester $s$ (GPA_Prior). The variable $RA_i^s$ is the roommate average GPA in semester $s$ (RM_Avg), $RD_i^s$ is the differences in roommate GPAs in semester $s$ (RM_Diff), and $RA_i^s \times RD_i^s$ is their interaction term. The variable $D^{Ge}$ is a gender dummy, which is coded as 1 and 0 for females and males, respectively. The variables $D^{Ma}, D^{Co}$, and $D^{Se}$ are major, cohort, and semester dummies, respectively (see Supplementary Table 3 for details).

Moreover, we employ an OLS model to study the relationship between a student's in-dorm ordinal rank (OR_InDorm) according to prior performance and their future performance after controlling their prior performance, the average and differences in roommate prior performance, their gender, major, cohort, and semester. Meanwhile, we examine how this relationship is moderated by other factors, including peer heterogeneity. Specifically, the empirical specification is given by

$$G_i^{s+1} = b_0 + b_1 G_i^s + b_2 OR_i^s + b_3 RA_i^s + b_4 RD_i^s + b_5 OR_i^s \times G_i^s + b_6 OR_i^s \times RA_i^s + b_7 OR_i^s \times RD_i^s + b_8 D^{Ge} + b_9 D^{Ma} + b_{10} D^{Co} + b_{11} D^{Se} + \epsilon_i, \tag{5}$$

where $OR_i^s$ is the OR_InDorm of student $i$ in semester $s$ (ranging from 1 to 4) and $\epsilon_i$ is the error term. The interaction terms are $OR_i^s \times G_i^s$ between OR_InDorm and GPA_Prior, $OR_i^s \times RA_i^s$ between OR_InDorm and RM_Avg, and $OR_i^s \times RD_i^s$ between OR_InDorm and RM_Diff for student $i$ in semester $s$. All other controls are the same as above (see Supplementary Information Section 5 for details on these variables and Supplementary Table 3 for summary statistics).

### Reporting summary

Further information on research design is available in the Nature Portfolio Reporting Summary linked to this article.

## Data availability

All data necessary to replicate the statistical analyses and main figures are available in Supplementary Information and have been deposited in the open-access repository Figshare (https://doi.org/10.6084/m9.figshare.25286017)[92]. The raw data of anonymized student accommodation and academic performance are protected by a data use agreement. Those who are interested in the raw data may contact the corresponding authors for access after obtaining Institutional Review Board (IRB) approval.

## Code availability

All code necessary to replicate the statistical analyses and main figures has been deposited in the open-access repository Figshare (https://doi.org/10.6084/m9.figshare.25286017)[92].

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

## Acknowledgements

The authors thank Min Nie, Shimin Cai, Defu Lian, Zhihai Rong, Huaxiu Yao, Yifan Wu, Lili Miao, and Linyan Zhang for their valuable discussions. This work was partially supported by the National Natural Science Foundation of China Grant Nos. 42361144718 and 11975071 (T.Z.) and the Ministry of Education of Humanities and Social Science Project Grant No. 21JZD055 (T.Z.).

## Author contributions

T.Z. and J.G. designed research; T.Z. collected data; Y.C. and J.G. performed research; Y.C., T.Z., and J.G. analyzed data; J.G. wrote the paper; Y.C. and T.Z. revised the paper.

## Competing interests

The authors declare no competing interests.
