## [Peer Review File · Nature Communications]

REVIEWER COMMENTS

Reviewer #1 (Remarks to the Author):

In this paper, the authors quantify roommate peer effects by developing a null-model approach to analyze the longitudinal data on student accommodation and academic performance from a university in China. They utilize the “almost” random assignment of roommates at the beginning to show that later in the following semesters there is similarity in the roommate GPAs something that is attributed to peer effects. They further explore peer effects through some OLS models. While the paper is well written, the study technically sounds and the topic is interesting in the broad area of peer effects and education, I am not convinced about the randomization that is the key assumption of the study. The authors have no access to the exact process of the random assignment neither have access to demographic data to check it themselves. While at the first read I was quite enthusiastic about the study and how cool and neat are the methods/results, I later developed some reservations regarding the random assignment process. For that reason, I am recommending rejection unless the authors can evaluate in some way the initial random assignment and further make sure that later in the semesters there is no significant selection process (i.e., move from one room to the other).

Some more comments:

- the model free identification of peer effects pick the relation of grades between roommates at the same semester, while the OLS approach try to related this semester grade with past semester peer's performance. This needs to be clear in the text.
- To show the robustness of the OLS result, a simple falsification test is to run the same regression on the roommate null model.

Reviewer #2 (Remarks to the Author):

In this manuscript, the authors propose a null model to quantify the effect of peer influence on academic performance among students sharing dorm rooms in Chinese Universities. The authors show that students tend to assort positively over time regarding their relative performance percentiles. Hence, the authors argue that they see evidence of a positive peer influence among students.

In general, the manuscript is well-written. Some methodological aspects could have been clear (e.g., the null model should be discussed in the main article). However, the study's goals are well-supported and clearly understood. This work constitutes an interesting, unique, and novel contribution worth publication. It is noteworthy to state that the novelty comes from the data (size, detail, and potential) rather than the approach used to identify the null model, which follows standard statistical methods.

I have some questions and comments that I would invite the authors to clarify.

1. Students are assigned randomly to their dorm rooms. What is unclear is whether these rooms are located in the same building/community or different ones. The results show that students sharing the same dorm room tend to assort positively over time, but is this mediated by the fact that they are all influenced by similar external factors? For instance, being located in wealthier neighborhoods, closer to the university (minimizing commute time), or inserted in communities with different cultures? Shouldn't the regressions include fixed effects for such groups if that is the case? Perhaps that is the case, but the authors don't have data to make this treatment. If this is a study limitation, they should at least discuss it.
2. The individuals in the dataset are all Chinese, or does it include foreign students?
3. The authors compare the averages of the data distribution with the null distribution obtained from bootstrap. A t-test shows that the distributions are different, but another way of looking at it would be to look at the significance of each assimilation observation. Which ones are significantly different from the null? And how many show values above average and significant according to the null? What characterizes such observations?
4. The null model could be augmented; for instance, have the authors ensure that the ensembles, although random, maintain the same composition of majors and other individual characteristics? Going back to point 1., if dorms are located in different locations, have the authors randomized students only within locations? This would naturally allow to control for additional effects
5. Is there any effect on gender? Is assimilation lower/greater among dorms of boys/girls? Or in dorms for mixed genders? If this is a study limitation, the authors should discuss such in light of the existing literature.
6. Do the authors agree that the effects, although positive, are marginal? Can the authors explain why that's the case?
7. How is the number of students per major distributed? Do we have majors with a low number of students? What is the likelihood that students of the same major get paired together?

Short comments

- "Contagions" is repeated in line 29/30

- In Line 34, I don't see how Smoking, Substance Abuse, or Alcoholism can be related to students' demotivation and academic performance. Please rephrase the sentence. The authors want to state that such behaviors are well-known examples of behaviors adopted/acquired through social influence, which can also be true for other behaviors associated with negative student performance.

- Line 51, please define "usually"

- The choice of words in lines 59/63 was poor. Regression models also contrast against a null model. It might not be the most suitable to disentangle non-linear behaviors. Moreover, null models in complex systems are arguably challenging due to the nature of the phenomena. But, more importantly, I would be careful to compare two methodologies at different maturity levels and look to achieve different purposes. Right now, what the authors wrote is suggestive that one is better than the other, and I don't think that's what the authors want to state.

- In line 79, remove the "and understand"

- I find using a "developing a null-model" approach is rather bleak and not informative. Null models are pervasive whenever one wants to assess the magnitude of an effect. The authors need to be more precise in their language.

- Line 81, what are the cohorts? How many majors?

- Line 116, the actual probability is the frequency of observations.

- Are all dorm rooms located at the same address?

- Since the authors claim that their null model is a big part of their novelty, I don't think delegating such to the methods is reasonable. Perhaps the authors should explain in detail in the body of the manuscript how the null model is estimated.

- Line 189 please correct the sentence "continues is significant"

- Line 203, I don't understand the sentence "... if a student is consistently ranked low in a dorm room even though the ranking among all students is high ... ". How can that be?

Reviewer #3 (Remarks to the Author):

This is an interesting paper building upon a rich tradition of social network interventions in university settings. I have major concerns with how the statistical tests insufficiently reflect the actual dormitory assignment process. Some additional details and hopefully helpful comments are found below.

Statistical Model

It is appropriate to take a null-model based approach to test the causal effect of randomizations on students' similarity (note my critique about the null model below). However, it should be clarified that this approach is not novel in the social networks literature. Such tests have been introduced as conditional uniform graph tests many decades ago. One recent application of such models in a similar context is the paper by Boda et al. (2020, Scientific Reports). When the authors write that "we develop a general null-model approach to demonstrate roommate peer effects on student academic performance" it may appear that the statistical method itself is new.

Interpretation of findings

I believe that the authors should focus not only on the statistical significance of their findings, but also on the effect sizes. How much more similar are room mates compared to their random counterparts? Is the difference meaningful?

True randomization and the null model differ

The paper suggests that they have a quasi-experimental setting due to the random room assignment. However, I think this is not the case. The causal arguments in the paper hinges on the random treatment, so I believe it is important to consider any non-random aspects. The authors write that "In order to live on campus, incoming new students have to fill out housing applications to the university housing management office and sign a housing contract with the office. Within the application, students have the opportunity to choose a dormitory and provide their lifestyles and living preferences. The university housing assignment team reviews each student's preferences and uses them for dorm and suite mate matching".

The null model should therefore replicate the exact assignment process. Otherwise the considered lifestyle and dormitory preferences will be different in the null model and in particular, students will expected to be more different than in the empirically observed network which is a confounding factor to the interpretation of (academic) similarity as being the result of social influence. The authors appear not control for preferences as it was done by the housing administration. The SI states that "Specifically, we start with the actual roommate configuration and randomly shuffle all students multiple times with the restriction of four roommates per dorm (see Supplementary Information Section 3 for details). We generate 1000 roommate null models and average them at the dorm room level to produce a plausible roommate null model, based on which we calculate the relative ratio and the assimilation measure." The preferences are not considered, which is a problem for the subsequent causal interpretation, as I mentioned above.

Interpretation of lasting ties

It is important to note that comparing the “lasting” ties is not a causal inference test based on the initial random assignment anymore. Some co-habitation ties may be shorter-lived as individuals decide to break those ties by moving out. These individual choices are likely not to be independent from similarity / homophily. This is unclear when the authors write: “assimilation exhibits an overall increasing trend over time, suggesting a stronger peer effect on academic performance as roommates live together longer.”. It reads like the authors interpret this increasing similarity between cohabitants in the long run as a causal effect of the treatment. This is inappropriate.

Code availability

I do not see a reason why the code was not made available for the review process already.

Point-by-Point Response

Reviewer #1:

In this paper, the authors quantify roommate peer effects by developing a null-model approach to analyze the longitudinal data on student accommodation and academic performance from a university in China. They utilize the “almost” random assignment of roommates at the beginning to show that later in the following semesters there is similarity in the roommate GPAs something that is attributed to peer effects. The further explore peer effects through some OLS models. While the paper is well written, the study technically sounds and the topic is interesting in the broad area of peer effects and education, I am not convinced about the randomization that is the key assumption of the study. The authors have no access to the exact process of the random assignment neither have access to demographic data to check it themselves. While at the first read I was quite enthusiastic about the study and how cool and neat are the methods/results, I later developed some reservations regarding the random assignment process. For that reason, I am recommending rejection unless the authors can evaluate in some way the initial random assignment and further make sure that later in the semesters there is no significant selection process (i.e., move from one room to the other).

Reply: We thank the Reviewer for this thoughtful review of our manuscript. We are delighted that the Reviewer found the topic interesting in the broad area of peer effects and education and was enthusiastic about the study. We very much appreciate the Reviewer’s insightful comments and valuable suggestions, which have helped us further clarify the study and strengthen its findings. In the following, we provide a point-by-point response and highlight our major changes.

Here the Reviewer raised a very insightful point that the initial random assignment of students to dorm rooms is important to this study. We completely agree that the random assignment process needs evaluation and more validation. The Reviewer’s question indeed pushed us to dive deeper into understanding the data generation process and improve its documentation. Following the Reviewer’s suggestions, we have spent considerable effort reviewing the exact process of the initial assignment and validating the randomization. In particular, we consulted several university administrators from the Housing Management Office and the Student Admission Office and yielded three pieces of evidence supporting the random assignment, which we provide below.

First, the initial assignment followed the university’s student accommodation management regulations, and there was no consideration of personal preferences. Before student registration in the first semester, the housing office received a list of incoming undergraduate students with the information of their name, gender, and student ID. Detailed departmental information is coded in the student ID (e.g., “2008060205003”, a hypothetical student ID to illustrate the structure; the same below), where the first four digits represent the cohort (e.g., “2008”, namely, the year of student admission), each of the following two digits represents the school (e.g., “06”: School of Computer Science and Engineering), major (e.g., “02”: Information Security major in the school), and administrative unit (e.g., “05”: the 5th unit in the major; one unit contains about 30 students), and the last three digits represent the student’s order in their administrative unit (e.g., “003”: the 3rd student in the unit). Upon receiving the student list, the housing office separated female and male students because there was no gender mix in dorm rooms or buildings and then assigned them to dorm rooms in order of student ID. There, students of the same gender, administrative unit, major, or school were assigned to the same 4-person dorm room, on the same floor, and in the

same dorm building when possible. For example, students “2008060205001,” “2008060205002,” “2008060205003,” and “2008060205004” were assigned together. When an administrative unit had remaining students, they were paired with students in the next unit in the same major or school when possible. Our analysis of the accommodation data showed that the student accommodation management regulations were enforced on the initial roommate assignment. Specifically, we found that roommates tend to have nearby student IDs (**Figure R1.1a**), a pattern that is more pronounced in actual data than in a roommate null model in which students are randomly shuffled within the same cohort, gender, and major (see SI Section 3.1 for details), and roommates in most dorm rooms have the same major (**Figure R1.1b**). Moreover, it was confirmed that the housing office neither had access to students’ prior academic performance or other personal data nor allowed them to choose roommates based on their personal preferences. Students were informed of their accommodation information only when they checked in before the first semester.

Figure R1.1: Some patterns of the initial roommate assignment. (a) Roommates tend to have nearby student IDs. In more than 60% of dorm rooms, at least three roommates have nearby student IDs in the actual data, while the fraction is less than 2% in the roommate null model. (b) Roommates tend to have the same major. In more than 90% of dorm rooms, four roommates have the same major. Because the roommate null model controls for major, the results are the same in the actual data and the null model.

Second, the initial assignment did not depend on the student’s prior academic performance. We consulted the admission office to understand the relationship between student IDs and academic performance before college enrollment. We were informed that there were no simple, one-size-fits-all criteria to compare the academic ability of students who were admitted into the same major. The assignment of student IDs was plausibly random for the following reasons. (1) The university used a quota system to admit students into different majors, and students were admitted in the order of majors to which they applied. The quotas for different provinces or regions were allocated according to the university’s discretion and preferences, whereas the university-hosting province received more quotas. (2) The National Higher Education Entrance Examination (i.e., “Gaokao” in China) was held by provincial governments using different evaluation systems. The total score of the exams was hardly comparable across different systems (not like the total SAT score in the US, for example). Therefore, no simple criteria were obvious enough for administrators to evaluate students’ academic performance before college enrollment. (3) The admission office randomly assigned students in the same major to one or more administrative units (about 30 students in each administrative unit) and assigned a student ID number to each student (see above) without considering their prior academic performance. For students in the same major, the only obvious consideration was gender. Specifically, in each administrative unit, male students were assigned

smaller student IDs (e.g., “2008060205003”) than female students (e.g., “2008060205029”). There was no mix of majors in the same administrative unit.

Third, there was no significant selection process later in the semesters. Once assigned to a dorm room by the housing office, four students lived together until graduation. There was no systematic reassignment of roommates or change of dorm rooms after the initial assignment. Moving out of a dorm room or from one dorm room to the other, as mentioned by the Reviewer, should be very rare on a few occasions. All applications for a dorm room or roommate change were reviewed by the housing office. Exceptions were only given in justified circumstances, such as when a student transferred to a different program, applied for sick leave and had irreconcilable conflicts with other roommates. According to the housing office, these circumstances were very rare at the time.

Taken together, the initial roommate assignment is plausibly random concerning students’ prior academic performance and preferences. While we lacked data on student prior performance and personal data to directly verify the initial random assignment, our consultation with university administrators gave us extra confidence in assuming that the initial assignment of roommates was random because the administrative processes involved no obvious consideration of student prior academic performance, socioeconomic background, or personal preference.

To further ensure that our roommate null model replicates the initial assignment process, we added additional controls for the same cohort, gender, and major during random shuffling of students (see SI Section 3.1), finding that our results are robust (see, for example, **Figure R1.2**). In the revision, we have added more details about the exact roommate assignment process (see SI Section 1.2) and discussed the limitations of the data in the main text (see Methods and Discussion). We feel that adding these details has improved the clarity of the study and its findings.

Figure R1.2: Assilimation of roommate academic performance. (a) The density distribution of assimilation for all dorm rooms. (b) The increasing actual assimilation relative to the null model.

Some more comments:

- the model free identification of peer effects pick the relation of grades between roommates at the same semester, while the OLS approach try to related this semester grade with past semester peer’s performance. This needs to be clear in the text.

Reply: Thanks for the insight and great suggestion. We have clarified the differences between the two analyses. Specifically, in the null-model analysis, we demonstrated roommate peer effects by

showing significantly larger assimilation of academic performance in the actual data than in the null model. There, we calculated the differences in the academic performance of roommates in the same semester and compared them with the null-model results. In the OLS regression, on the other hand, we conducted a Granger causality type of analysis to show that a student’s future academic performance is indeed impacted by the average academic performance of their roommates in the past. This prior-post data construction helps address concerns about reverse causality. In the revision, we have added more explanations (see Results and SI Section 5).

- To show the robustness of the OLS result, a simple falsification test is to run the same regression on the roommate null model.

Reply: We appreciate this insightful suggestion, which inspired us to further study the robustness of the OLS regression results. Following the Reviewer’s guidance, we conducted a falsification test by running the same OLS regressions on the roommate null model, where students are randomly shuffled within the same cohort, gender, and major (see SI Section 3.1 for details). Once assigned to the same room in the null model, students are assumed to live together for all semesters.

Figure R1.3: Results of the falsification test for the effects of roommate average prior performance based on the roommate null model. (a) Full regression results. (b) Main effects. (c) Moderation effects.

Analyzing the regression results based on the roommate null model consisting of 100 independent implementations of the random shuffling process, we found that the effect of roommate average prior performance (RM_Avg) on a student’s future performance (GPA_Post) is very small ($b = 0.003$ in the null model vs 0.042 in the actual data), and the moderation effect of the differences in roommate prior performance (RM_Diff) is also small ($b = -0.011$ in the null model vs -0.089 in the actual data; **Figure R1.3**). Moreover, the effect of a student’s within-dorm ordinal rank according to prior performance (OR_InDorm) on the student’s future performance (GPA_Post) is very small ($b = 0.001$ in the null model vs 0.006 in the actual data), and the moderating effect of RM_Diff is also small ($b = -0.007$ in the null model vs -0.022 in the actual data). Taken together, these regression results from the falsification test support our findings that roommate peer effects on performance are nontrivial, significant, and robust. In the revision, we have included these results in SI (see SI Section 5.3) and added more discussions to the main text (see Results).

Overall, we are very grateful for all the insightful feedback from the Reviewer. Please feel free to let us know if there’s anything else we could do to further improve the paper. Thank you!

Reviewer #2:

In this manuscript, the authors propose a null model to quantify the effect of peer influence on academic performance among students sharing dorm rooms in Chinese Universities. The authors show that students tend to assort positively over time regarding their relative performance percentiles. Hence, the authors argue that they see evidence of a positive peer influence among students.

Reply: We thank the Reviewer for this excellent summary of the study and all these thoughtful comments and suggestions, which helped us further strengthen the work. We have benefited tremendously from the Reviewer's valuable input. Below, we provide a point-by-point response and highlight major changes that we made in the revision.

In general, the manuscript is well-written. Some methodological aspects could have been clear (e.g., the null model should be discussed in the main article). However, the study's goals are well-supported and clearly understood. This work constitutes an interesting, unique, and novel contribution worth publication. It is noteworthy to state that the novelty comes from the data (size, detail, and potential) rather than the approach used to identify the null model, which follows standard statistical methods.

Reply: We appreciate the Reviewer's insights and encouragement. We are delighted that the Reviewer recognized the work's contribution and helped clarify it further. We completely agree with the Reviewer that, while standard statistical methods were widely used to study peer effects, the data analyzed in the study is valuable and has led to new findings. Indeed, utilizing statistical methods, in particular the roommate null model, we developed an assimilation metric to assess the magnitude of roommate peer effects on academic performance. Also, the new longitudinal dataset of student accommodation and academic performance allowed us to understand how a student is impacted by their roommates and the effects of peer heterogeneity. Following the Reviewer's suggestion, in the revision, we have clarified the novelty, discussed the roommate null model in the main article (see Results), and improved the methodological aspects (see Methods).

I have some questions and comments that I would invite the authors to clarify.

1. Students are assigned randomly to their dorm rooms. What is unclear is whether these rooms are located in the same building/community or different ones. The results show that students sharing the same dorm room tend to assort positively over time, but is this mediated by the fact that they are all influenced by similar external factors? For instance, being located in wealthier neighborhoods, closer to the university (minimizing commute time), or inserted in communities with different cultures? Shouldn't the regressions include fixed effects for such groups if that is the case? Perhaps that is the case, but the authors don't have data to make this treatment. If this is a study limitation, they should at least discuss it.

Reply: The Reviewer raised an excellent point regarding the geographic location of dorm rooms in the study. We appreciate this insight. Different from Western universities, where on-campus student residential buildings are integrated into local communities, universities in China have clear geographical boundaries protected by fences, and gates are guarded. In our study, dorm buildings are all within the university boundary (i.e., on campus), and they are identical and geographically very close (see **Figure R2.1** for a sketch of the campus facility layout). Per the university's student accommodation management regulations, students of the same gender, major, or school were

randomly assigned to each 4-person dorm room on the same floor in the same dorm building when possible or in nearby buildings depending on capacity (see SI Section 1.2 for details). This suggests that the initial assignment of on-campus dorm rooms is also plausibly random concerning the dorm environment. While we agree with the Reviewer that external factors may mediate the assimilation of roommate academic performance, our analysis can't rule out the factors outside of the dorm environment. In the revision, we have further discussed this limitation (see Discussion).

Figure R2.1: A sketch of the campus facility layout of the university under study.

Following the Reviewer's suggestion on regression, we have further added major fixed effects to the OLS regression, which are stronger controls than dorm buildings because students in the same major live in the same building when possible, and one building usually hosts one or more majors as per the building's capacity. We show that our regression results remain robust (see Table 1 and Table 2 in the main text). In the revision, we have added more details about the dorm rooms under study (see SI Section 1.2) and further discussed the study's limitations (see Discussions).

2. The individuals in the dataset are all Chinese, or does it include foreign students?

Reply: The sample only covers Chinese students, and it doesn't include foreign students. In the revision, we have clarified this (see Methods and SI Section 1.3).

3. The authors compare the averages of the data distribution with the null distribution obtained from bootstrap. A t-test shows that the distributions are different, but another way of looking at it would be to look at the significance of each assimilation observation. Which ones are significantly different from the null? And how many show values above average and significant according to the null? What characterizes such observations?

Reply: We appreciate the Reviewer's inspiring suggestions on exploiting the data in another way. Following the Reviewer's guidance, we further examine the significance of each dorm room's assimilation. Specifically, we first calculate the average assimilation (mean) of all dorm rooms under the roommate null model consisting of 100 independent implementations (see below and SI Section 3.1 for details) and its standard error (*s.e.*). We then calculate the assimilation of each dorm room and identify those that have significantly larger assimilation than the average null-model assimilation (P value = 0.05; a value that is larger than $\text{mean} + 1.96 \times \text{s.e.}$, i.e., the upper 95% CI).

We find that about 60% of dorm rooms have larger-than-null-model assimilation, and this fraction exhibits an increasing trend (**Figure R2.2**), suggesting that peer effects overall increase as students live together longer. Our conclusions are now supported by two different methods. In the revision, we have added these results together with some discussions (see Results and SI Section 4.3).

Figure R2.2: The assimilation of roommate academic performance and the temporal trend. (a) The share of dorm rooms with larger-than-null-model assimilation according to the 95% CI. (b) Results by semesters.

4. The null model could be augmented; for instance, have the authors ensure that the ensembles, although random, maintain the same composition of majors and other individual characteristics? Going back to point 1., if dorms are located in different locations, have the authors randomized students only within locations? This would naturally allow to control for additional effects.

Reply: We appreciate the Reviewer’s insights into the null model construction. Following the Reviewer’s suggestion, we have augmented the null model by adding more constraints to the random shuffling process. Specifically, we now only allow random shuffles of students in the same cohort, gender, and major so that the ensembles maintain the same composition of controlled student characteristics. Considering that dorm rooms are located in geographically close buildings and students in the same major live together when possible (see above for details), we don’t additionally control for locations. With the augmented null model, we redid all the analyses, finding that the results are robust (see, for example, **Figure R2.3**). In the revision, we have updated the results and added more details about the null model (see Results and SI Section 3.1).

Figure R2.3: Assimilation of roommate academic performance. (a) The density distribution of assimilation for all dorm rooms. (b) The increasing actual assimilation relative to the null model.

5. Is there any effect on gender? Is assimilation lower/greater among dorms of boys/girls? Or in dorms for mixed genders? If this is a study limitation, the authors should discuss such in light of the existing literature.

Reply: The Reviewer raised another excellent question regarding the effect on gender. In the study, female and male students were assigned to dorm rooms in different buildings, and there were no gender-mixed dorm rooms or dorm buildings. Thanks to the Reviewer’s question, we realized we could further unpack the analysis by gender. We find that, while females represent only a small proportion of the student sample (about 23%), they have significantly better academic performance on average than their male counterparts (P value < 0.001 ; **Figure R2.4a**), which is consistent with the positive coefficient of the gender dummy (where females and males are coded as 1 and 0, respectively; namely, females are in the treatment group) in Table 1 and Table 2 of the main text. Yet, we find that there is no significant gender difference in the average assimilation of roommate academic performance either for all five semesters together (P value > 0.1 ; **Figure R2.4b**) or for each semester (P value > 0.1 ; **Figure R2.4c**). In the revision, we have added these new analyses (see SI Section 4.4) and briefly discussed them (see Results and Discussion).

Figure R2.4: The roommate peer effects for female and male students. (a) The average GPA for five semesters. (b) The average assimilation for five semesters. (c) The average assimilation for each semester.

6. Do the authors agree that the effects, although positive, are marginal? Can the authors explain why that’s the case?

Reply: We fully agree with the Reviewer that while the roommate peer effects are positive and statistically significant, the effect size appears to be modest. Specifically, based on our roommate null model analysis, the mean actual assimilation (0.549) of all dorm rooms is about 10.7% larger than the mean null-model assimilation (0.496; **Figure R2.3**). Moreover, based on our regression analysis, a 100-point increase in roommate average prior GPA percentile is associated with a 6-point increase in the student’s GPA percentile (see Table 1 in the main text). The effect is about 7% as large as the effect of a 100-point increase in the focal student’s prior GPA percentile, and it is about 12% of the student’s own GPA percentile. Inspired by the Reviewer’s question, we reviewed the literature and found that prior studies reported effect sizes at a similar scale for various environments (e.g., dormitories and classrooms) and cultures (e.g., Western universities). For instance, Sacerdote (2001) found that a 1-*s.d.* increase in the roommate’s GPA is associated with a 0.05 increase in the student’s freshmen year GPA (an effect size of about 4% of their own GPA), Zimmerman (2003) found that a 100-point increase in roommate verbal SAT score

translated into a 0.03 increase in the student’s first-semester GPA (an effect size of about 15% as large as a 100-point increase in their own verbal SAT score), Carman et al. (2012) found that a 0.1-*s.d.* increase in peer average math achievement increases the student’s math test score by 0.04 *s.d.*, and Thiemann (2021) reported negative results that 1-*s.d.* increase in average peer ability is associated with a decrease in a student’s GPA by 7.9% of 1-*s.d.* on average. Presumably, many other external factors may contribute to this observation of modest effect size, such as that dorm rooms are not formal learning environments like classrooms and libraries, and there are other channels through which students learn from each other. In the revision, we have expanded the discussions on the effect size together with prior works (see Results and SI Section 5.1).

7. How is the number of students per major distributed? Do we have majors with a low number of students? What is the likelihood that students of the same major get paired together?

Reply: Thanks for these questions. In the study, we inferred the major of each student from their student ID. We dropped those for which we couldn’t draw a confident inference or that had missing values in control variables (see SI Section 1.3 for details). We find that most majors have 55~150 students, and majors with a few students are rare (**Figure R2.5a**). One major has fifteen students in the sample at the minimum. The likelihood that students of the same major were paired is very high. Specifically, in over 90% of dorm rooms, roommates are in the same major (**Figure R2.5b**). It should be noted that per the university’s student accommodation management regulations, the roommate assignment according to student ID will naturally result in a few mixed-major dorm rooms due to the remaining 1~3 students in each major (see SI Section 1.2 for details). In the revision, we have added these statistics and discussions (see Methods and SI Section 1.3).

Figure R2.5: Student compositions in each major or dorm room. (a) Distribution of majors by the number of students per major. (b) Share of dorm rooms by the number of same-major students in the dorm room.

Short comments

- “Contagions” is repeated in line 29/30

Reply: Fixed.

- In Line 34, I don’t see how Smoking, Substance Abuse, or Alcoholism can be related to students’ demotivation and academic performance. Please rephrase the sentence. The authors want to state that such behaviors are well-known examples of behaviors adopted/acquired through social

influence, which can also be true for other behaviors associated with negative student performance.

Reply: We have rephrased the sentence following the Reviewer’s suggestion. Now it reads “some well-known examples of human behaviors adopted through social influence, such as smoking, substance abuse, and alcohol use, are often associated with negative student performance.”

- Line 51, please define “usually”

Reply: We have revised the sentence to better explain the assignment process. Now it reads “most Chinese universities randomly assign students to dorm rooms”. We have also added more details about the assignment process to the main text (see Methods) and SI (see SI Section 1.2).

- The choice of words in lines 59/63 was poor. Regression models also contrast against a null model. It might not be the most suitable to disentangle non-linear behaviors. Moreover, null models in complex systems are arguably challenging due to the nature of the phenomena. But, more importantly, I would be careful to compare two methodologies at different maturity levels and look to achieve different purposes. Right now, what the authors wrote is suggestive that one is better than the other, and I don’t think that’s what the authors want to state.

Reply: Thanks for these helpful comments and suggestions. In our study, we used both null models and regression models to study roommate peer effects on academic performance. Specifically, in the null-model analysis, we demonstrated the presence of roommate peer effect and assessed their magnitude by comparing the assimilation of roommates’ academic performance in the same semester in the actual data with that in the null model. In the regression analysis, on the other hand, we conducted a Granger causality type of analysis to show that a student’s future academic performance is impacted by the average academic performance of their roommates in the past. This prior-post data construction helps better address concerns about reverse causality. While each of these two methodologies has its unique advantages and can achieve different purposes, our study benefits from using both to demonstrate the robustness of our conclusions.

We completely agree with the Reviewer that regression models also contrast against a null model. Also, null models are not new in social science studies, and they have been used to study peer effects in complex social systems (see, for example, Raudenbush 2002 and Boda *et al.* 2020). Building on traditional statistical methods, our hypothesis test using the roommate null model is indeed a permutation test (one-sided), which is often referred to as the “quadratic assignment procedure” in social network studies (see, for example, Hubert & Schultz 1976 and Krackhardt 1988). In the revision, we have revised the narrative and introductions about the two widely used methodologies in social science studies, acknowledging their common grounds and highlighting their unique strengths in studying peer effects (see Introduction and Discussion).

- In line 79, remove the “and understand”

Reply: Fixed.

- I find using a “developing a null-model” approach is rather bleak and not informative. Null models are pervasive whenever one wants to assess the magnitude of an effect. The authors need to be more precise in their language.

Reply: We have revised the sentence to make the language more precise. Now it reads “we quantify roommate peer effects using both null models and regression approaches”.

- Line 81, what are the cohorts? How many majors?

Reply: The same cohort of students here means those who were admitted by the university in the same year. There are 36 majors for the two cohorts of students under study (34 majors for one cohort and 31 majors for the other). In the revision, we have added more details about the cohorts and majors (see Results, Methods, SI Section 1.2, and SI Section 1.3).

- Line 116, the actual probability is the frequency of observations.

Reply: Fixed.

- Are all dorm rooms located at the same address?

Reply: Yes. All dorm rooms and dorm buildings are on-campus, located in the same campus area, and very close to each other in distance (see above and SI Section 1.1 for details).

- Since the authors claim that their null model is a big part of their novelty, I don't think delegating such to the methods is reasonable. Perhaps the authors should explain in detail in the body of the manuscript how the null model is estimated.

Reply: We appreciate this insightful comment. In the revision, we have added more details about the roommate null model, and how we use it to demonstrate the presence of roommate peer effects on academic performance and further assess its magnitude (see Results). Meanwhile, we have clarified that our approaches (both null models and regression models) follow standard statistical methods and, as suggested by the Reviewer, the novelty primarily comes from the longitudinal dataset of student accommodation and academic performance (see Results and SI Section 1.3).

- Line 189 please correct the sentence “continues is significant”

Reply: Fixed.

- Line 203, I don't understand the sentence “... if a student is consistently ranked low in a dorm room even though the ranking among all students is high ... “. How can that be?

Reply: Thanks for catching this. In a dorm room that contains four students all with high academic performance (e.g., with a GPA percentile ≥ 0.9 among students in the same major), there is a possibility that one student's ordinal rank in the dorm room is consistently 4th across different semesters (e.g., with a GPA percentile = 0.9, which is smaller than their three roommates' GPA percentile), although this may not be common in the actual data. In the revision, we have revised the sentence and added more explanations to avoid confusion (see Results and SI Section 5.2).

Overall, we are very grateful for all the insightful feedback from the Reviewer. Please feel free to let us know if there's anything else we could do to further improve the paper. Thank you!

Reviewer #3:

This is an interesting paper building upon a rich tradition of social network interventions in university settings. I have major concerns with how the statistical tests insufficiently reflect the actual dormitory assignment process. Some additional details and hopefully helpful comments are found below.

Reply: We appreciate the Reviewer's helpful review of the manuscript. We are delighted that the Reviewer found the study interesting, and we are grateful for the Reviewer's comments, which have helped us to strengthen the study. In the following, we provide a point-by-point response and highlight the major changes we have made in the revision.

Statistical Model

It is appropriate to take a null-model based approach to test the causal effect of randomizations on students' similarity (note my critique about the null model below). However, it should be clarified that this approach is not novel in the social networks literature. Such tests have been introduced as conditional uniform graph tests many decades ago. One recent application of such models in a similar context is the paper by Boda et al. (2020, Scientific Reports). When the authors write that "we develop a general null-model approach to demonstrate roommate peer effects on student academic performance" it may appear that the statistical method itself is new.

Reply: Thanks for these insights and for suggesting the very relevant literature. We completely agree with the Reviewer that null-model approaches are not new and they have been applied to study peer effects in the social sciences. As pointed out by the Reviewer, applications of null models in a similar context have emerged in the social network literature. In particular, the paper by Boda et al. (2020) is a recent example of prior works, which used randomizations to study the impact of short-term network interventions on long-term social relationships between students.

In the revision, we have revised the narrative to clarify that our approaches build on traditional statistical methods (see Results). More specifically, inspired by permutation tests, often referred to as the "quadratic assignment procedure" in social network studies (see, for example, Hubert & Schultz 1976 and Krackhardt 1988), we perform a statistical hypothesis test to check whether the assimilation of roommates' academic performance in the actual data deviates significantly from its theoretical value that is proxied by the value in the roommate null model. Moreover, we have added more discussions of the relevant literature to recognize prior works that used null models in studying peer effects in complex social systems (see Introduction and Methods).

Interpretation of findings

I believe that the authors should focus not only on the statistical significance of their findings, but also on the effect sizes. How much more similar are room mates compared to their random counterparts? Is the difference meaningful?

Reply: We thank the Reviewer for these valuable suggestions. We agree with the Reviewer that we should also discuss the effect sizes as they are important for interpreting the results. Indeed, we developed an assimilation metric based on the roommate null model to quantify the extent to which the academic performance of roommates is more similar than expected by chance. We found that roommates are on average about 10.7% more similar in their academic performance than their

random counterparts (mean assimilation 0.549 in the actual data vs 0.496 in the null model), and this difference is statistically significant (**Figure R3.1**). Our regression analyses further suggested that, after controlling for a student’s prior academic performance and other factors including cohort, gender, major, and semester, a 100-point increase in the roommate’s average prior GPA percentile is associated with a 6-point increase in the focal student’s future GPA percentile. This effect is about 7% as large as the effect of the student’s prior GPA percentile, and it is about 12% of the student’s future GPA percentile (see Table 1 in the main text). Although the difference appears to be modest (see below for details), it is still meaningful considering the significant explanatory power of roommate prior performance for the focal student’s future performance.

Figure R3.1: The assimilation of roommate academic performance. (a) The density distribution of assimilation for all dorm rooms. The upper and lower halves show the actual and null-model assimilation, respectively. (b) The overall increasing actual assimilation from semester 1 to semester 5. The y-axis shows the percentage difference between the mean actual assimilation and the mean null-model assimilation.

The effect sizes we found in the study are at a similar scale as those reported in the literature for various environments (e.g., dormitories and classrooms) and cultures (e.g., Western universities). For instance, Sacerdote (2001) found that a 1-*s.d.* increase in the roommate’s GPA is associated with a 0.05 increase in the student’s freshmen year GPA (an effect size of about 4% of their own GPA), Zimmerman (2003) found that a 100-point increase in roommate verbal SAT score translated into a 0.03 increase in the student’s first-semester GPA (an effect size of about 15% as large as a 100-point increase in their own verbal SAT score), Carman et al. (2012) found that a 0.1-*s.d.* increase in peer average math achievement increases the student’s math test score by 0.04 *s.d.*, and Thiemann (2021) reported negative results that 1-*s.d.* increase in average peer ability is associated with a decrease in a student’s GPA by 7.9% of 1-*s.d.* on average. In the revision, we have expanded the effect size analysis and added more discussions (see Results and Discussion).

True randomization and the null model differ

The paper suggests that they have a quasi-experimental setting due to the random room assignment. However, I think this is not the case. The causal arguments in the paper hinges on the random treatment, so I believe it is important to consider any non-random aspects. The authors write that “In order to live on campus, incoming new students have to fill out housing applications to the university housing management office and sign a housing contract with the office. Within

the application, students have the opportunity to choose a dormitory and provide their lifestyles and living preferences. The university housing assignment team reviews each student's preferences and uses them for dorm and suite mate matching”.

Reply: We completely agree with the Reviewer that it is important to consider any non-random aspects of the initial assignment of roommates. The sentences mentioned by the Reviewer describe the assignment in Western universities rather than in Chinese universities. To avoid confusion, we have revised this paragraph in the SI to clarify the content (see below and SI Section 1.2).

“In Western countries, most universities provide on-campus dormitories to undergraduate students [...]. To live on campus, incoming new students have to fill out housing applications to the university housing management office and sign a housing contract with the office. [...] The university housing assignment team reviews each student's preferences and uses them for dorm and suite mate matching. If students provide no such preferences, [...] students with similar lifestyles and living preferences have a higher probability of living together.”

We are particularly grateful for the Reviewer’s insight into the discrepancy between the roommate null model and the true randomization of roommate assignments in the Chinese university under study. Inspired by the Reviewer’s comment, we have spent considerable effort in evaluating the initial random assignment and figuring out any non-random aspects by consulting the university’s administrators and analyzing the actual accommodation data. We found several pieces of evidence supporting that the initial assignment of roommates involved no obvious consideration of students’ prior academic performance, socioeconomic background, or personal preference (see below and SI Section 1.2 for details about the exact initial assignment process). Moreover, there was no significant reassignment of roommates or individual selections later in the semesters.

At the same time, our consultations revealed some non-random aspects of the initial roommate assignment that are not directly related to academic performance. Specifically, students in the same major and cohort tended to live in the same dorm room when possible, and there was no gender-mixed dorm room or building. To better account for these non-random aspects, we have augmented the roommate null model by adding more constraints to the random shuffling process. Specifically, we now only allow random shuffles among students in the same cohort, gender, and major. These controls are in line with the true randomization, and the ensembles of dorm rooms in the null model preserve the composition of controlled student characteristics (see SI Section 3.1 for details). We show that our main conclusions are robust when using this augmented roommate null model to perform the analysis (see **Figure R3.1**, Results, SI Section 4, and SI Section 5).

Taken together, we have clarified the roommate assignment process in Chinese and Western universities, added detailed documentation of the initial assignment process, and augmented the roommate null model. We feel that these revisions have improved the clarity of the study.

The null model should therefore replicate the exact assignment process. Otherwise the considered lifestyle and dormitory preferences will be different in the null model and in particular, students will expected to be more different than in the empirically observed network which is a confounding factor to the interpretation of (academic) similarity as being the result of social influence. The authors appear not control for preferences as it was done by the housing administration. The SI states that “Specifically, we start with the actual roommate configuration and randomly shuffle

all students multiple times with the restriction of four roommates per dorm (see Supplementary Information Section 3 for details). We generate 1000 roommate null models and average them at the dorm room level to produce a plausible roommate null model, based on which we calculate the relative ratio and the assimilation measure.” The preferences are not considered, which is a problem for the subsequent causal interpretation, as I mentioned above.

Reply: We completely agree with the Reviewer that ideally we would want the null model to replicate the exact initial assignment process. The Reviewer’s question indeed pushed us to dive deeper into understanding the data generation process and improve its documentation in great detail. In the following, we first introduce more details about the exact initial assignment process and then explain how our revised roommate null model replicates this process.

We yielded three pieces of evidence supporting that the initial assignment is plausible random after consulting several university administrators from the Housing Management Office and the Student Admission Office. First, the initial assignment followed the university’s student accommodation regulations, and there was no consideration of student preferences. Before student registration in the first semester, the housing office received a list of incoming students with the information of their name, gender, and student ID. Detailed departmental information is coded in the student ID (e.g., “2008060205003”, a hypothetical student ID to illustrate the structure; the same below), where the first four digits represent cohort (e.g., “2008”, namely, the year of admission), each of the following two digits represents the school (e.g., “06”: School of Computer Science and Engineering), major (e.g., “02”: Information Security major in the school), and administrative unit (e.g., “05”: the 5th unit in the major; one unit contains about 30 students), and the last three digits represent the student’s order in their administrative unit (e.g., “003”: the 3rd student in the unit). Upon receiving the student list, the housing office separated female and male students because there was no gender mix in dorm rooms and then assigned them to dorm rooms in order of student ID. There, students of the same gender, administrative unit, major, or school were assigned to the same 4-person dorm room, on the same floor, and in the same dorm building when possible. For example, those with student IDs “2008060205001,” “2008060205002,” “2008060205003,” and “2008060205004” were assigned together in one dorm room. When an administrative unit had remaining students, they were paired with students in the next unit in the same major or school.

Our analysis of the accommodation data showed that the university’s student accommodation management regulations were enforced on the initial assignment. Specifically, we found that roommates tend to have nearby student IDs (**Figure R3.2a**), a pattern that is more pronounced in the actual data than in the roommate null model in which students are shuffled randomly within the same cohort, gender, and major (see SI Section 3.1 for details), and roommates in most dorm rooms have the same major (**Figure R3.2b**). Moreover, it was confirmed that the housing office neither had access to students’ prior academic performance or other personal data nor allowed them to choose roommates based on their personal preferences. Students were informed of their accommodation information only when they checked in before the first semester.

Second, the initial assignment did not depend on the student’s prior academic performance. We consulted the admission office to understand the relationship between student IDs and academic performance before college enrollment. We were informed that there were no simple, one-size-fits-all criteria to compare the academic ability of students who were admitted into the same major. The assignment of student IDs was plausibly random for the following reasons. (1) The university

used a quota system to admit students into different majors, and students were admitted in the order of majors to which they applied. The quotas for different provinces or regions were allocated according to the university’s discretion and preferences, whereas the university-hosting province received more quotas. (2) The National Higher Education Entrance Examination (i.e., “Gaokao” in China) was held by provincial governments using different evaluation systems. The total score of the exams was hardly comparable across different systems (not like the total SAT score in the US, for example). Therefore, no simple criteria were obvious enough for administrators to evaluate students’ academic performance before college enrollment. (3) The admission office randomly assigned students in the same major to one or more administrative units (about 30 students in each unit) and assigned a student ID number to each student (see above) without considering their prior academic performance. For students in the same major, the only obvious consideration was gender. Specifically, in each unit, male students were assigned smaller student IDs (e.g., “2008060205003”) than female students (e.g., “2008060205029”). There was no mix of majors in the same unit.

Figure R3.2: Some patterns of the initial roommate assignment. (a) Roommates tend to have nearby student IDs. In more than 60% of dorm rooms, at least three roommates have nearby student IDs in the actual data, while the fraction is less than 2% in the roommate null model. (b) Roommates tend to have the same major. Because the roommate null model controls for major, the results are the same in the actual data and the null model. In more than 90% of dorm rooms, four roommates have the same major.

Third, there was no significant selection process later in the semesters. Once assigned to a dorm room by the housing office, four students lived together until graduation. There was no systematic reassignment of roommates or change of dorm rooms after the initial assignment. Moving out of a dorm room or from one room to the other should be very rare on a few occasions. All applications for a dorm room or roommate change were reviewed by the housing office. Exceptions were only given in justified circumstances, such as when a student transferred to a different program, applied for sick leave and had irreconcilable conflicts with other roommates. According to the housing office, these circumstances were very rare at the time. Taken together, these three pieces of evidence support that the initial assignment is random concerning students’ academic performance, socioeconomic background, or personal preference.

The Reviewer’s comments helped us realize that we should further augment the roommate null model to ensure that it replicates the initial assignment process. To this end, we added several additional controls to the random shuffling of students (see SI Section 3.1 for details). We didn’t control for lifestyle and dormitory preferences because these were not considered in the initial

assignment as mentioned above. Specifically, to construct a roommate null model, we start with the actual roommate configuration and randomly shuffle students between dorm rooms while preserving their compositions of cohort, gender, and major (see, for example, **Figure R3.2b**, which suggests that the roommate null model preserves the composition of roommate majors in the actual data). By repeating this random shuffling process, we construct a plausible roommate null model that consists of 1000 independent implementations (see SI Section 3.1 for details). We find that our conclusions remain robust when using the augmented roommate null model to perform the analysis (see, for example, **Figure R3.1**). In the revision, we have added more details about the initial assignment process and the roommate null model (see Methods and SI Section 3.1). We feel that adding these details has improved the clarity of the study.

Interpretation of lasting ties

It is important to note that comparing the “lasting” ties is not a causal inference test based on the initial random assignment anymore. Some co-habitation ties may be shorter-lived as individuals decide to break those ties by moving out. These individual choices are likely not to be independent from similarity / homophily. This is unclear when the authors write: “assimilation exhibits an overall increasing trend over time, suggesting a stronger peer effect on academic performance as roommates live together longer.”. It reads like the authors interpret this increasing similarity between cohabitants in the long run as a causal effect of the treatment. This is inappropriate.

Reply: We appreciate the Reviewer’s insights and helpful comments, which prompted us to further clarify the roommate relationship in the study. Indeed, the establishment and maintenance of the roommate relationship is different from a one-time intervention at the beginning of the first semester. Here, roommates lived together throughout the undergraduate program after being assigned to a dorm room before the first semester. There was no systematic reassignment after the initial assignment. Moving out or from one room to the other was very rare on a few occasions (see above and SI Section 1.2 for details). In other words, there was no significant individual choice later in the semesters. For the period during which student academic performance was measured, cohabitation ties (i.e., roommate relationships) remained stable and active, instead of being “shorter-lived” ties or “lasting” ties that were first established but then broken due to personal choices. As we mentioned in the main text (see Introduction and Discussion), this is one strength of our study using the longitudinal data of student accommodation and academic performance to study student peer effects. Presumably, these stable cohabitation ties become stronger the longer roommates live together. Therefore, we interpret the increasing assimilation of roommates’ academic performance as evidence of stronger roommate peer effects over time. In the revision, we have revised the sentence and clarified the roommate relationship under study to avoid confusion (see Results, Methods, and SI Section 1.2).

Code availability

I do not see a reason why the code was not made available for the review process already.

Reply: We originally submitted the code as a zip file. We have now deposited the code into a public repository and made it fully available (see the “code availability” section in the main text).

Overall, we are very grateful for all the insightful feedback from the Reviewer. Please feel free to let us know if there’s anything else we could do to further improve the paper. Thank you!

REVIEWERS' COMMENTS

Reviewer #1 (Remarks to the Author):

In this paper, the authors quantify roommate peer effects by developing a null-model approach to analyze the longitudinal data on student accommodation and academic performance from a university in China. They utilize the “almost” random assignment of roommates at the beginning to show that later in the following semesters there is similarity in the roommate GPAs something that is attributed to peer effects. The further explore peer effects through some OLS models. While the paper is well written, the study technically sounds and the topic is interesting in the broad area of peer effects and education.

During the first round of reviews I was not convinced about the random assignment -- the key assumption of the study. However, further investigation/analysis by the authors during the review process has shown that the assignment is somehow random. Therefore, I would recommend publication.

Reviewer #2 (Remarks to the Author):

[no comments to the Author available]

Point-by-Point Response

Reviewer #1:

In this paper, the authors quantify roommate peer effects by developing a null-model approach to analyze the longitudinal data on student accommodation and academic performance from a university in China. They utilize the “almost” random assignment of roommates at the beginning to show that later in the following semesters there is similarity in the roommate GPAs something that is attributed to peer effects. The further explore peer effects through some OLS models. While the paper is well written, the study technically sounds and the topic is interesting in the broad area of peer effects and education.

During the first round of reviews I was not convinced about the random assignment -- the key assumption of the study. However, further investigation/analysis by the authors during the review process has shown that the assignment is somehow random. Therefore, I would recommend publication.

Reply: We thank the Reviewer for reviewing our manuscript and providing an excellent summary of the key results. We are delighted that the Reviewer found the study technically sound and recommended the paper for publication.

Reviewer #2:

[no comments to the Author available]

Reply: We thank the Reviewer for reviewing our manuscript.